# Multisensory Integration in *Caenorhabditis elegans* in Comparison to Mammals

**DOI:** 10.3390/brainsci12101368

**Published:** 2022-10-09

**Authors:** Yanxun V. Yu, Weikang Xue, Yuanhua Chen

**Affiliations:** 1Department of Neurology, Medical Research Institute, Zhongnan Hospital of Wuhan University, Wuhan University, Wuhan 430070, China; 2Frontier Science Center for Immunology and Metabolism, Wuhan University, Wuhan 430070, China

**Keywords:** multisensory integration, *Caenorhabditis elegans*, sensory processing, sensory modality, sensory input, neural plasticity

## Abstract

Multisensory integration refers to sensory inputs from different sensory modalities being processed simultaneously to produce a unitary output. Surrounded by stimuli from multiple modalities, animals utilize multisensory integration to form a coherent and robust representation of the complex environment. Even though multisensory integration is fundamentally essential for animal life, our understanding of the underlying mechanisms, especially at the molecular, synaptic and circuit levels, remains poorly understood. The study of sensory perception in *Caenorhabditis elegans* has begun to fill this gap. We have gained a considerable amount of insight into the general principles of sensory neurobiology owing to *C. elegans*’ highly sensitive perceptions, relatively simple nervous system, ample genetic tools and completely mapped neural connectome. Many interesting paradigms of multisensory integration have been characterized in *C. elegans*, for which input convergence occurs at the sensory neuron or the interneuron level. In this narrative review, we describe some representative cases of multisensory integration in *C. elegans*, summarize the underlying mechanisms and compare them with those in mammalian systems. Despite the differences, we believe *C. elegans* is able to provide unique insights into how processing and integrating multisensory inputs can generate flexible and adaptive behaviors. With the emergence of whole brain imaging, the ability of *C. elegans* to monitor nearly the entire nervous system may be crucial for understanding the function of the brain as a whole.

## 1. General Introduction

Multisensory integration is an essential issue in the fields of cognition, behavioral science and neurobiology. It studies how information from different modalities, such as sight, sound, smell, taste and touch, becomes integrated as a coherently meaningful representation in the nervous system [1]. Successful integration can decrease sensory uncertainty and reaction latency and form better memory and perception [1], thus providing adaptive advantages for survival and reproduction.

Although sensory processing was traditionally viewed and studied in modality-specific manners, different regions of the mammalian brain are enormously interactional. Numerous studies have identified multisensory neurons in cortical areas that were previously classified as uni-sensory [2]. Multisensory integration is probably necessary for almost all animal activities. Ample evidence demonstrates that multisensory inputs are commonly found in many ascending pathways [2,3]. This leads to researchers proposing that “the entire cortex (brain?) is multisensory” [1,2,4,5,6], albeit the functional roles of the integration have not all been characterized.

There are two well-accepted principles of multisensory integration: the temporal and spatial principle and the inverse effectiveness principle [2,7,8,9]. The spatial and temporal principle states that integration is more likely to happen or be strengthened when the uni-sensory stimuli occur at approximately the same location or close in time. The principle of inverse effectiveness states that the magnitude of integration is inversely related to the responsiveness of individual stimuli, i.e., weak stimuli presented in isolation are more likely to elicit or strengthen multisensory integration [9,10,11].

The ability to integrate cross-modal senses is not inherent. Instead, it develops gradually after birth with the presence of cross-modal events in the environment. Even though multisensory neurons are produced early in life, these neurons cannot integrate multisensory inputs until much later [12]. This capability gradually matures into young adulthood. Therefore, multisensory integration is essentially a learned ability, involving the neural mechanism of plasticity.

Multisensory processing appears to be disrupted in several neuropsychiatric disorders, including autism spectrum disorder, dyslexia, attention deficit hyperactivity disorder, sensory processing disorder and schizophrenia [13,14,15,16,17,18]. How multisensory processing relates to these disorders is still unclear. It has been shown that multisensory training can restore visual function in visual cortex-damaged animals [2]. In some cases of autism, the delayed development of multisensory processing can be improved with age, presumably via prolonged development [19]. Since sensory integration intervention is based on neural plasticity [20], this gives hope that individually tailored multisensory training techniques can ameliorate these neuropsychiatric disorders with multisensory processing deficits.

*Caenorhabditis elegans* (*C. elegans*) senses its complex environment using multisensory integration strategies to make behavioral decisions [21,22]. Studies of multisensory integration in *C. elegans* have a unique niche due to the intrinsic properties of this organism’s nervous system. There are many advantages to studying neurobiology in *C. elegans*, which can be extended to the study of multisensory integration. *C. elegans* has a well-defined and compact nervous system with 302 neurons and it is the only organism whose entire neuronal connectome is mapped throughout different developmental stages [23,24,25]. Recently, the worm “contactome” has also been mapped, adding spatial context to the connectome [26,27]. In addition, gene expression profiles at single cell resolution of all 302 neurons have been generated [28,29].

Moreover, numerous genetic tools for neuronal functional studies have been developed in *C. elegans*. A single or a few neurons can be selectively killed by laser ablation [30], by expressing caspase to induce apoptosis [31], or by using miniSOG, a photosensitizer generating singlet oxygen [32,33] in a cell type-specific manner. Neuronal activity can be manipulated opto-genetically [34] or chemo-genetically [35]. Those tools greatly facilitate the identification of an underlying neural circuit. Moreover, there is an arsenal of worm mutants in various membrane potential-associated proteins, synaptic and signaling proteins, along with the ease of generating transgenic and knock-out animals, facilitating the investigation of molecular functions of the nervous system.

Previous studies in this field have revealed substantial mechanisms of sensory integration at the molecular, cellular, synaptic and circuit level in *C. elegans*. There are two excellent reviews [21,22] summarizing sensory processing circuits and sensory integration paradigms in *C. elegans*. In this narrative review, we will compare multisensory integration processing in mammals and *C. elegans* with a focus on *C. elegans*, concentrating on new paradigms that have not been covered before. Using representative examples and easy-to-relate comparisons, we hope this essay will help colleagues investigating sensory processing in mammals to comprehend and appreciate the research in *C. elegans*.

## 2. Multisensory Integration in *C. elegans*

### 2.1. Sensory Processing in C. elegans

*C. elegans* has 60 sensory neurons that can sense a variety of sensory modalities, including smell, taste, touch, temperature, light, color, oxygen, CO_2_, humidity, proprioception, magnetic field and sound [36,37,38,39,40,41,42,43,44,45]. For each environmental stimulus assayed in isolation, the fundamental neural circuit is well characterized [28] and the corresponding behavioral output is generally robust.

Worms use diverse protein receptors to sense environmental stimuli. The *C. elegans* genome encodes over 1000 predicted G protein-coupled receptors (GPCRs), many of which are likely to function as receptors in sensory neurons [37]. The one confirmed odorant receptor is ODR-10, which detects diacetyl [46]. GPCR LITE-1 has been shown to be a photoreceptor [47]. It has been demonstrated that the receptor guanylyl cyclase GCY-35 is an oxygen sensor [48]. Several receptor guanylyl cyclases and a glutamate receptor have been proposed as thermo-receptors [49,50]. The mechano-sensor is thought to be made up of two ion channel subunits, MEC-4 and MEC-10, from the degenerin/epithelial Na+ channel (DEG/ENaC) family [51,52].

When the GPCR protein receptors are activated by a stimulus, the signal is transduced by two types of downstream ion channels [37,38]. One type consists of the TRP (transient receptor potential) channels, OSM-9 and OCR-2 [53,54]. The other type of downstream signaling transduction is mediated by the second messenger cGMP, involving receptor guanylyl cyclases and cyclic nucleotide-gated channels TAX-4 and TAX-2 [55,56]. Both types of channels can mobilize calcium, open voltage-gated calcium channels and activate the sensory neuron.

The organization of the sensory system from all modalities is vastly different in *C. elegans* compared to mammals due to its numerical simplicity. Take the olfactory sensory neurons, for example. In *C. elegans*, a pair of each AWA, AWB and AWC neurons serve as the primary odorant chemosensory neurons, while worms are likely to express around 100 GPCRs as presumed odorant receptors [57]. Therefore, each odorant-sensing neuron expresses many receptors. This is in contrast to the “one neuron, one receptor” rule in mammals, which refers to the fact that each olfactory sensory neuron expresses one and only one olfactory receptor [58]. In the ascending pathways beyond the sensory neuron layer, the sensory systems in mammals are much more complex. Their projections travel a long distance and project to multiple higher brain regions. In *C. elegans*, interneurons comprise the largest group of neurons, which is probably the counterpart of the higher brain regions in mammals [24]. They can be divided into first-layer, second-layer and commander interneurons. Sensory neurons project to different layers of interneurons and converge into five commander interneurons that control muscle movement [59].

### 2.2. C. elegans Performs Multisensory Integration

All animals, including lower organisms such as *C. elegans,* can integrate information from multiple channels to form an accurate presentation of the complex environment. The integration process allows animals to make better choices based on the information they have received. The environment of *C. elegans* may contain both beneficial elements such as mates and food, but also harmful elements such as poison and predators. How to integrate environmental cues in a context-dependent manner and make an appropriate decision is a central theme in the studies of *C. elegans* neurobiology. Despite having just 60 sensory neurons, *C. elegans* exhibits an array of highly sensitive sensory modalities and displays diverse paradigms of multisensory integration [21,22]. These paradigms can probably be divided into two categories: (1) exposing *C. elegans* to two sensory modalities of opposing valence and studying how worms make decisions; (2) exposing *C. elegans* to stimuli from two sensory modalities and examining how the behavior evoked by one stimulus is altered by a second stimulus. All the paradigms found in *C. elegans* seem to be consistent in that multisensory integration can change perception.

Processing various sensory inputs at the level of sensory neurons or sensilla in the periphery is one way to accomplish multisensory integration. It can also be accomplished by integrating at the interneuron or central nervous system levels. In addition, an animal’s internal state and past experiences can top-down alter the output of sensory-evoked behavior. Below is a detailed discussion of *C. elegans*’ integration paradigms and top-down mechanisms.

Theoretically, two stimuli from the same sensory modality, for example, two different odorants, can also interact with each other. This scenario does not seem to be included in studies of multisensory integration in mammals but is often studied in *C. elegans*, providing many interesting sensory integration paradigms. In evolution, sensory integration from the same modality is likely to be fundamental to sensory integration from multiple modalities [12]. It has been found that low concentrations of different odorants often have a synergistic effect in mice [60]. This is reminiscent of the principle of inverse effectiveness. Therefore, some paradigms demonstrating sensory integration from the same modality in *C. elegans* will also be discussed below.

### 2.3. Integration at the Level of Sensory Neurons

Many organisms contain polymodal sensory neurons, meaning that those neurons can each sense multiple stimuli from different sensory modalities. In that case, polymodal sensory neurons can easily integrate sensory information from different modalities. Although sensory neurons are highly specialized in mammals, polymodal sensory neurons do exist, as exemplified by cutaneous C-fiber nociceptors [61,62]. They can respond to more than one type of noxious stimuli applied to the skin, usually mechanical, chemical and thermal [61,62]. Studying these polymodal nociceptors has provided great significance in pain management [63].

Many sensory neurons in *C. elegans* are polymodal. For example, the ASH neuron pair is the main nociceptor sensory neuron, which mediates avoidance responses to noxious stimuli [37]. It can sense an array of aversive cues, such as high osmolality, quinine, nose touch, repellent chemicals, heavy metals, and so on. Interestingly, after ASH activation, *C. elegans* can separately process stimuli from different modalities by innovating different downstream postsynaptic receptors [64]. Although high osmolality and nose touch both activate ASH neurons, high osmolality utilizes both non-NMDA and NMDA receptor subunits to mediate the avoidance response, whereas nose touch only triggers non-NMDA receptors post-synaptically [64,65]. Genetic and electrophysiological analysis suggests that this modality-specific signal transduction is because high osmolality enables increased glutamate released from ASH neurons, which is sufficient to activate both non-NMDA and NMDA receptors [65].

In addition to ASH, many other sensory neurons in *C. elegans* are also polymodal. For example, the chemosensory AWC neuron pair can respond to temperature [66,67]. Similarly, the AFD neuron pair primarily senses temperature but can also respond to CO_2_ [68,69]. These polymodal neurons all have the ability to mediate multisensory integration (Figure 1A).

In mammalian studies, multisensory integration is generally referred to as integration that occurs at the level of the sensory cortex or higher, which is beyond the first synapse in an ascending pathway [12]. Nonetheless, polymodal sensory neurons are an efficient way for stimuli from multiple modalities to be integrated through facilitation or inhibition.

### 2.4. Integration at the Level of Interneurons

Multisensory encoding in mammals takes place in many higher brain regions, such as the superior colliculus (SC) in the midbrain and many regions in the cerebral cortex [6,70]. Due to the significant restriction on the number of neurons, *C. elegans* often encodes the valance of a stimulus at the sensory neuron level [71]. Nonetheless, many paradigms of multisensory integration occur at the first- and second-layer interneurons to modulate the sensory output.

The hub-and-spoke circuit is a well-known sensory integration paradigm. One of these regulates the worm’s social behavior, or aggregation. In this circuit, the interneuron RMG acts as the hub, linking to multiple sensory neurons (the spokes) with gap junctions [72]. High activity in the RMG is essential for promoting social aggregation, of which the activity level can be modulated by several spoke neurons that sense diverse stimuli, including oxygen, sex pheromones and noxious chemicals (Figure 1B). This circuit connection motif integrates cross-modal sensory inputs to ensure a coherent output. Another similar hub-and-spoke circuit regulates nose touch response [73,74,75]. This involves the interneuron RIH being the hub connecting to sensory neurons ASH, FLP and OLQ responding to gentle touch via gap junctions.

Other interneurons can also serve as the node in a circuit. Interneuron AIA can receive inputs from many chemosensory neurons. AIA receives excitatory input from an electrical synapse and disinhibitory inputs via chemical synapses [76]. The two types of inputs need to happen coincidently to improve the reliability of AIA’s response [76]. The logic of this integrating neuron seems to relate closely to the temporal principle of multisensory integration.

Recently, a two-layer integration has been reported to modulate foraging behavior in *C. elegans* [77]. Forage is a stereotyped local search behavior looking for food. The behavior requires redundant inhibitory inputs from two interneuron pairs, AIA and ADE, which receive chemosensory and mechanosensory food-related cues, respectively [77]. Sensory cues symbolizing food are first organized into the chemosensory cues that are integrated at AIA and the mechanosensory cues that are integrated at ADE. Input from these two neurons subsequently integrates into the next layer of interneurons. Local search behavior can be triggered when either of these two sensory cues is removed (Figure 1C).

### 2.5. Neuromodulators in Multisensory Integration

In mammals, neuromodulators such as monoamines and neuropeptides play an important role in regulating brain states and sensory integration [78]. Due to their widespread projections and slow action, neuromodulators can shape neural activity in many locations across multiple time scales. Neuromodulators can modulate a wide range of behaviors in *C. elegans*, including multisensory integration [79].

Tyramine [80,81], insulin-like peptides [82], serotonin [83], octopamine [84] and dopamine [84] have all been implicated in regulating multisensory integration in *C. elegans*. The tryptophan-kynurenine metabolic pathway has been associated with a variety of neurodegenerative and psychiatric disorders [85,86,87]. Kynurenic acid, a metabolite in this pathway, is depleted during fasting, leading to activation of interneuron, thus regulating food-dependent behavioral plasticity in *C. elegans* [88].

### 2.6. Top-Down Mechanisms in the Multisensory Integration

Sensory information transduction is thought to follow through a hierarchy of brain areas that are progressively more complex. “Top-down” refers to the influences of complex information from higher brain regions that shapes early sensory processing steps. Top-down influences can affect sensory processing at all cortical and thalamic levels [89]. Common top-down modulators of sensory processing can include stress, attention, expectation, emotion, motivation and learned experience [89,90,91,92].

Although *C. elegans* lacks cognition and emotion, the sensory output can be influenced by its past experience and internal physiological states, such as hunger and sickness. The most well-studied top-down modulator in *C. elegans* is probably starvation, likely to be due to a lack of other top-down cognitive or emotional modulators. Hunger will increase *C. elegans*’ preference for seeking attractive odors cueing for food availability in the risk of other harmful stimuli [81,93,94].

In a risk-reward choice assay [81], *C. elegans* is trapped inside a circle of a repulsive hyperosmotic fructose solution, while an attractive food odor is placed outside the circle. The outcome is scored on whether worms cross the aversive circle to reach the attractive odor. Almost no worms would exit the circle in the initial 15 min. However, after being starved for 5 h, almost 80% of the worms would exit the repulsive circle, seeking the attractive odor. The interneuron RIM is identified as modulating this decision via a top-down extra-synaptic aminergic signal [81]. In another scenario of multisensory integration between opposing valences, the insulin/IGF-1 signaling (IIS) pathway is mediating the signal of hunger to decrease responses to the repellent gustatory cue [94]. Several other neuromodulators have also been found to relay the signal of starvation to functionally reconfigure sensory processing and, presumably, they can also mediate top-down regulation impinging upon multisensory integration.

Past experience is another well-studied top-down modulator for sensory processing in *C. elegans*. A recent study demonstrated how worms can learn to navigate a T-maze to locate food via multisensory cues [95]. In general, past experience affects sensory processing via reshaping the synapse. Here, we provide two examples to demonstrate how prior experience can change either the strength or the composition of the synapse to enable plasticity. *C. elegans* does not have an innately preferred temperature. Instead, it remembers its cultivation temperature and moves to that temperature when subjected to a temperature gradient [96]. This sensory memory is encoded by the synaptic strength between the thermo-sensory neuron pair AFD and its downstream interneuron AIY [97]. Under warmer temperatures, this synapse is strengthened, enabling worms to move to warmth and vice versa. Similarly, *C. elegans* cultivated at a certain NaCl concentration can remember this concentration and travel to it when subjected to a NaCl gradient [98]. This gustatory memory is encoded by differentially innervating the glutamate receptors in the AIB neuron, which is postsynaptic to the salt-sensing neuron ASE right (ASER). At a higher salt cultivation condition, decreasing NaCl concentration causes ASER activation, triggers glutamate released from ASER and subsequently activates the excitatory glutamate receptor GLR-1 in the downstream AIB neurons, whereas, cultivated in a lower salt environment, glutamate released from ASER activates the inhibitory glutamate receptor AVR-14 in AIB instead [99].

## 3. Multisensory Integration in Development

In mammals, the ability to perform multisensory integration is not an inherent ability. Even in the newborn rhesus monkey, who can already see and hear very well at birth, neurons in the SC cannot integrate coincident cross-modal sensory stimuli [100]. Its emergence requires cross-modal experience in a way that seems to optimize the animal’s survival in the environment it is exposed to [12]. Not much is known about the mechanism driving multisensory integration in development [101].

Parallel studies are lacking in *C. elegans* with only a few sensory-related studies looking at sensory processing across development. Recent publications find that juvenile worms have different behaviors [102,103] and neural wiring [25] from adults. The difference in food-seeking behavior seems to rise from the differently recruited olfactory neural circuits at different developmental stages [102].

Multisensory integration in development, aging and diseases is an important theme in mammalian studies. The loss of multisensory integration is also an indicator of neural function regression in the elderly population [104,105,106]. In the past, most studies in *C. elegans* neurobiology utilized young adults to avoid confounding from development and frailty in the elderly. Nonetheless, researchers have nowadays started to become interested in sensory processing in *C. elegans* across development and aging. With its powerful genetics, established neuronal connectome and accumulated knowledge in neurobiology, we believe *C. elegans* can continue to provide insights into the new field.

## 4. Comparison of Multisensory Integration Studies between *C. elegans* and Mammals

Despite their distance in evolution, mammals and *C. elegans* share some similarities in the principles of multisensory neurons. In terms of the organizing principle, many interneurons in *C. elegans* each receive inputs from different sensory modalities, which is reminiscent of the overlapping receptive fields in mammalian multisensory neurons. From many paradigms of sensory processing discussed here and elsewhere, many of the *C. elegans* interneurons are suitable for multisensory integration. A recurring theme in sensory processing in both *C. elegans* and mammals is that neuromodulators, especially monoamines, are involved in many paradigms of multisensory integration.

However, due to intrinsic differences between species, the technologies available and the varied study foci, there are significant disparities in multisensory integration research between *C. elegans* and mammals (Table 1). For example, when it comes to studying top-down mechanisms of multisensory integration in *C. elegans*, hunger is mostly used as the modulator, since influence from stress, attention, expectation, emotion, or motivation is not accessible in the lower organisms. There are other differences, to our knowledge, which are included below.

The major sensory modality in most mammals is vision. Therefore, many multisensory integration paradigms pair vision with a second stimulus from audio, somatosensory, or vestibular input. The major sensory modality in *C. elegans* is probably olfaction, so olfaction is most commonly paired with input from another modality such as taste, touch, temperature, oxygen, and so on. With the advancement of technology, methods to deliver spatially, temporally and quantitatively controlled stimuli in combination are emerging [107].

True multisensory integration does not seem to be tested strictly in *C. elegans*. In mammals, the fact that multisensory neurons are able to receive inputs from multiple modalities does not necessarily lead to multisensory integration. After successful integration, the magnitude of response from the multisensory neuron should be greater than the sum of the uni-sensory responses combined [1]. Therefore, whether to integrate or segregate simultaneous detected sensory signals during multisensory processing is a focus in mammalian studies.

Because true integration does not always happen, the spatial and temporal principle emphasizes that integration is more likely to happen or be strengthened when the uni-sensory stimuli occur at approximately the same location or close in time. Direct testing of this principle is challenging in *C. elegans* due to the limitations of the stimulus delivery method. Moreover, single neuron electrophysiological methods can be difficult in *C. elegans* due to the neurons’ small size [108]. The commonly implemented GECI (genetically encoded calcium indicators) for examining neuron activity comes only with a compromised resolution. The above makes it challenging to evaluate the individual neuron’s enhanced response to sensory integration.

Nonetheless, temporal integration is probably highly likely to happen because neuronal activity is rather slow in *C. elegans*. Action potentials are not readily visible in *C. elegans* neurons, which instead only display gradual neuronal activity [109,110]. These slow neuronal dynamics enables sensory integration to happen over a long period of time. It has been demonstrated that some behaviors in *C. elegans* require stimuli from two separate modalities working together [36], which indicates a remarkable amplification from true multisensory integration.

When multisensory integration takes place, many studies in *C. elegans* focus on its benefit for making a better decision based on more information inputs, so it is beneficial for survival. However, whether the decision is indeed beneficial is not tested. In mammals, multisensory integration has an array of readouts; it can increase response magnitude, reduce response latency, form more solid memories and generate more accurate perception. There is also a limited repertoire of behaviors that can be measured in *C. elegans*. Therefore, the behavior readout is often related to its movement or directed behaviors testing for the populational preference. This ties well with the research in *C. elegans*, which focuses on how worms make decisions.

The major advantages of using *C. elegans* for the study of neurobiology stem from its well-characterized neuronal connectome, ample molecular genetics tools to ablate, silence, or activate neurons and molecular tools facilitating the discovery of molecular mechanisms. From the examples listed here and in other *C. elegans* reviews, one can see that, in a sensory processing paradigm, detailed underlying mechanisms, including the composition of the neural circuits, the nature of synaptic connections, synaptic components and key signaling molecules, can all be discovered, which is still very hard to do in mammals at the current stage.

## 5. Conclusions

Multisensory integration is a fundamental issue in neurobiology and it has been explored mainly in mammalian systems. Relevant studies using *C. elegans* can offer unique advantages and have generated important insights into the understanding of sensory processing, including multisensory integration.

In the future, we anticipate *C. elegans* to continue to contribute to the research in multisensory integration with the newly developed multi-neuron imaging technique, in addition to its completely mapped neural circuits and powerful genetics. Nowadays, with the advancement of imaging technologies, large-scale brain activity recordings have become possible [111]. These technologies enable us to assess neural activity across the entire nervous system rather than examining neurons in isolation, which is especially important for studying multisensory processing since it can monitor many related neural regions simultaneously. However, current microscopy techniques are still unable to capture the activity of all the neurons across a functional network in the mammalian brain [112,113]. *C. elegans* is the only organism that can achieve single neuron resolution imaging during whole-brain activity recording and behavior [114,115]. We anticipate that using brain-wide neural activity recordings in conjunction with new theoretical approaches to interpret these data, as well as new optical [116] and synthetic approaches [117] in *C. elegans*, will allow scientists to understand the relationship linking sensory neural input and behavioral output, leading to a critical understanding in the field of multisensory integration.

## Figures and Tables

**Figure 1 brainsci-12-01368-f001:**
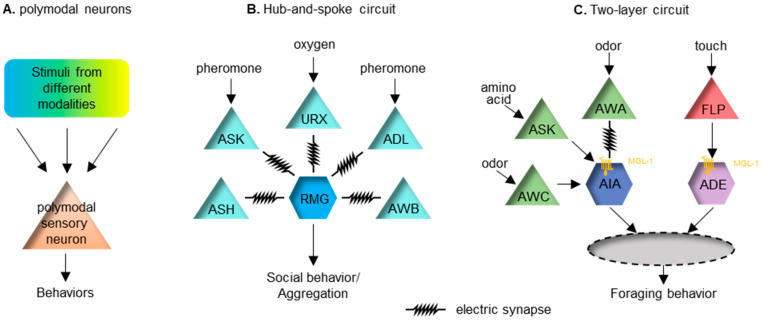
Several paradigms of multisensory integration in *C. elegans*. (**A**) Polymodal sensory neurons can receive and integrate inputs from different modalities. (**B**) A hub-and-spoke circuit. The hub neuron RMG is connected with pheromone-sensing neurons ASK and ADL, the oxygen-sensing neuron URX and several other sensory neurons via gap junctions. This circuit can integrate sensory inputs from and regulate social or aggregation behavior in *C. elegans*. (**C**) A two-layer circuit. Food-related chemosensory cues and mechanosensory cues are first integrated in parallel at the interneuron AIA and ADE, respectively, through the inhibitory metabotropic glutamate receptor MGL-1 (as symbolized by a yellow transmembrane protein), expressed post-synatpically in AIA and ADE. Additionally, glutamate can activate inhibitory ionotropic glutamate receptors in AIA. Signals from AIA and ADE will converge at the next level of the neural circuit to regulate foraging behavior in *C. elegans*.

**Table 1 brainsci-12-01368-t001:** Some differences comparing multisensory integration paradigms in *C. elegans* and mammals (see main context for details).

	*C. elegans*	Mammals
Dominant modality	Olfaction	Vision
Receptor expression	One neuron, many receptors	One neuron, one receptor
Valence of stimulus	Often determined at the sensory neuron level	Often determined at higher brain region such as amygdala
Common method measuring neural activity	Calcium imaging	Electrophysiology
Type of neuron membrane potential	Mostly graded potential	Action potential
Behavioral output	Often presented as directed behaviors, that involves a directional response to a directional sensory input	Presented as increased response magnitude, reduced response latency, more solid memories formation, more accurate perception and so on
Top-down modulators	Hunger is mostly used	Stress, attention, expectation, emotion, motivation and so on

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
