# Peer review of "Multisensory Integration in *Caenorhabditis elegans* in Comparison to Mammals"

_brainsci, 2022, doi:10.3390/brainsci12101368_

Round 1
Reviewer 1 Report
All in all, an interesting review.
I have a wish for the authors.
It is necessary to make a comparison table or figure for chapter 4. COMPARISON OF MULTISENSORY INTEGRATION STUDIES BETWEEN C. ELEGANS AND MAMMALS
Author Response
All in all, an interesting review.
I have a wish for the authors.
It is necessary to make a comparison table or figure for chapter 4. COMPARISON OF MULTISENSORY INTEGRATION STUDIES BETWEEN C. ELEGANS AND MAMMALS
We thank the reviewer for his/her interest and constructive comments. To address this issue, we have included a comparison table in the revised manuscript.

Reviewer 2 Report
The article by Yu et al. explains multisensory integration in C. elegans and then attempts to draw parallels to the systems of mammals. The topic may be of sufficient interest to researchers who study worms or mammals, as well as a broader audience. However, the article requires substantial revisions:
1. The authors must provide a more comprehensive overview of sensory and nervous systems in C. elegans and highlight similarities with mammalian sensory systems early in the review to emphasize the significance of this comparison.
2. At the conclusion of each section in which the authors cite examples of multisensory integration in C. elegans, the authors must specify whether or not such a system exists in mammals.
3. A goal of the study appears to be to highlight the recurring themes in multisensory integration in C. elegans and mammals, thereby supporting the use of C. elegans as a surrogate for mammalian systems. The authors must clarify this in the discussion system. They must identify similar multisensory integrations in nematodes and mammals and suggest how studying nematodes can provide insight into the mammalian system. They should also mention studies in which similar efforts were made, or the absence thereof.
4. 4. The authors must ensure that they provide references for every piece of information they present. There are numerous uncited sentences of this type throughout the text.
Author Response
The article by Yu et al. explains multisensory integration in C. elegans and then attempts to draw parallels to the systems of mammals. The topic may be of sufficient interest to researchers who study worms or mammals, as well as a broader audience. However, the article requires substantial revisions:
We thank the reviewer for his/her interest and constructive comments. Our responses to each point are below.
- The authors must provide a more comprehensive overview of sensory and nervous systems in C. elegansand highlight similarities with mammalian sensory systems early in the review to emphasize the significance of this comparison.
We have now added an overview of sensory processing in C. elegans to strengthen the background. In addition, we have streamlined the entire manuscript and made significant revisions in a number of places.
- At the conclusion of each section in which the authors cite examples of multisensory integration in C. elegans, the authors must specify whether or not such a system exists in mammals.
Unfortunately, there are more differences than similarity in multisensory integration studies in C. elegans and mammals at the current stage (as alluded to in the manuscript). Many examples in C. elegans are actually not found in mammals due to the vast differences between these two organisms (also see answer for the next point). In this regard, we are afraid that there is no direct point-to-point comparison of cases of multisensory integration in C. elegans and mammals.
- A goal of the study appears to be to highlight the recurring themes in multisensory integration in C. elegansand mammals, thereby supporting the use of C. elegansas a surrogate for mammalian systems. The authors must clarify this in the discussion system. They must identify similar multisensory integrations in nematodes and mammals and suggest how studying nematodes can provide insight into the mammalian system. They should also mention studies in which similar efforts were made, or the absence thereof.
Although we agree with the reviewer that C. elegans and mouse studies share some similarities, we do not think C. elegans can be a surrogate for mammalian systems. They each have their own set of advantages and disadvantages. We think the model organism C. elegans can provide insights into some aspects of multisensory integration study that may be hard to do in mammalian systems, especially at the molecular and synaptic levels. For example, plasticity is a recurring theme in multisensory integration. We have described in detail how exactly plasticity can occur in sensory processing in C. elegans by modulating synaptic strength or changing post-synaptic receptor types under different circumstances, and such detailed studies are lacking in mammals. Nonetheless, we have included a comparison table and rewritten many parts of this manuscript to more directly address “how studying nematodes can provide insight into the mammalian system”.
- 4. The authors must ensure that they provide references for every piece of information they present. There are numerous uncited sentences of this type throughout the text.
We have now added a lot more references, so that every statement has a reference(s).

Reviewer 3 Report
7 September 2022
Manuscript ID: brainsci-1918905
Type: Review
Title: ‘Multisensory Integration in C. elegans with Comparison to Mammals’ by Yu YV et al., submitted to Brain Sciences
Dear Authors,
The present review entitled ‘Multisensory Integration in C. elegans with Comparison to Mammals’ is a well-written and useful summary of the status of knowledge on the study of sensory perception in C. elegans. For this purpose, the authors selected some relevant evidence that focused on general principles of sensory neurobiology based on C. elegans’ highly sensitive perceptions and relatively simple nervous system. Results showed that many paradigms of multisensory integration have been characterized in C. elegans, for which input convergence occurs at the sensory neuron or the inter-neuron level. The authors concluded by stating that this manuscript describes some representative cases of multisensory integration in C. elegans, summarizing the underlying mechanisms, and comparing them with those in mammalian systems.
The main strength of this manuscript is that it addresses an interesting and timely question, describing how studies using C. elegans have generated important insights into the understanding of sensory processing, including multisensory integration. In general, I think the idea of this review is really interesting and the authors’ fascinating observations on this timely topic may be of interest to the readers of Brain Sciences. However, some comments, as well as some crucial evidence that should be included to support the author’s argumentation, needed to be addressed to improve the quality of the manuscript, its adequacy, and its readability prior to the publication in the present form, in particular reshaping parts of the introduction and discussion sections by adding more evidence and theoretical constructs.
Please consider the following comments:
1. Abstract: Please expand the abstract to 200 words, proportionally presenting the background, the rationale and the purpose of this review, the short summary, and the conclusion with the potential of this review.
2. Keywords: The keywords are missing. Please list ten keywords and use as many as possible in the first two sentences of the abstract.
3. A graphical abstract is highly recommended.
4. I would ask the authors to clarify the criteria they decided to use for studies’ collection in their review: they should specify the number of studies included in the review and the requirements used to decide whether a study met the inclusion/exclusion criteria of the review; they also should provide a more detailed description of all other variables for which data were sought and briefly present results of all statistical syntheses conducted. If the authors intend to present a narrative review, please state so in the abstract and the introduction.
5. The objectives of this study are generally clear and to the point; however, I believe that there are some ambiguous points that require clarification or refining. I think that authors here need to be explicit regarding how they operationally investigated and explored mechanisms of sensory processing in mammalian brain, since this is the key aim of this review.
6. Introduction: In the first-time the authors use the abbreviation ‘C. elegans’ in the text, they should present both the spelled-out version (Caenorhabditis elegans) and the short form.
7. Top-down mechanisms in the multisensory integration: In this paragraph, the authors described top-down mechanisms that regulate sensory integration. Since it may be challenging to readers to fully understand that the process of multisensory integration is not uniform, what factors do affect multisensory integration, and how the mammalian brain reconstructs a multisensory world at different states, I was wondering whether the author could be more specific and provide a brief overview of top-down factors that can modulate sensory processing and discuss their potential roles in multisensory integration, including any reference to support their statements.
8. Discussion: In my opinion, this review would be more compelling and useful to a broad readership if the authors moved beyond and discussed theoretical and methodological avenues in need of refinement, using this evidence to suggest a path forward. In this regard, I believe that it would have been essential to explore to use C. elegans as a model organism to study neurodegenerative diseases. In this regard, I would suggest adding evidence from recent studies that investigated hallmarks of neurodegeneration, like aggregates called “Lewy bodies” and the loss of dopaminergic neurons in frontal lobes (doi:10.17219/acem/139572; doi:10.1111/psyp.14122), which showed how overexpressing wild-type and mutated α-synuclein in dopaminergic neurons in these worms resulted in dopaminergic neuron loss. Also, in my opinion, it would be interesting to consider how human genes responsible for a range of mitochondrial diseases have an orthologous gene in preclinical models such as C. elegans, giving the possibility of using C. elegans as a model organism for studying mitochondrial diseases (https://doi.org/10.3390/cells11162607).
9. In my opinion, I think the ‘Conclusions’ paragraph would benefit from some thoughtful as well as in-depth considerations by the authors, because as it stands, it is very descriptive but not enough theoretical as a discussion should be. The authors should make an effort, trying to explain the theoretical implication as well as the translational application of their research, stating the potential of this review, the goal, the challenge, the knowledge or the technology necessary to achieve this goal, and future research direction, among others.
10. In according to the previous comment, I would ask the authors to include a ‘Limitations and future directions’ section before the end of the manuscript, in which authors can describe in detail and report all the technical issues brought to the surface.
11. Figure: According to the Journal’s guidelines, please provide a short explanatory caption for Figure 1.
12. References: I suggest presenting more references to present readers the solid background of this review manuscript. A review paper like this typically needs more than 150 references.
Overall, the manuscript contains 1 figure and 63 references. I believe that the manuscript might carry important value describing how studies using C. elegans have generated important insights into the understanding of sensory processing, including multisensory integration.
I hope that, after these careful revisions, this paper can meet the Journal’s high standards for publication.
I am available for a new round of revision of this paper. I declare no conflict of interest regarding this review.
Best regards,
Reviewer
Best regards,
Reviewer
Author Response
We thank the reviewer for his/her interest and constructive comments. Our responses to each point are below.
- Abstract: Please expand the abstract to 200 words, proportionally presenting the background, the rationale and the purpose of this review, the short summary, and the conclusion with the potential of this review.
We have now slightly expanded the abstract.
- Keywords: The keywords are missing. Please list ten keywords and use as many as possible in the first two sentences of the abstract.
Keywords are added and have been used more intensively in the abstract.
- A graphical abstractis highly recommended.
We feel a graphical abstract is more suitable for a research article describing new findings. Nonetheless, we have included a comparison table in the revised manuscript.
- I would ask the authors to clarify the criteria they decided to use for studies’ collection in their review: they should specify the number of studies included in the review and the requirements used to decide whether a study met the inclusion/exclusion criteria of the review; they also should provide a more detailed description of all other variables for which data were sought and briefly present results of all statistical syntheses conducted. If the authors intend to present a narrative review, please state so in the abstract and the introduction.
We have addressed this issue by stating that it is a “narrative review” both in the abstract and introduction.
- The objectives of this studyare generally clear and to the point; however, I believe that there are some ambiguous points that require clarification or refining. I think that authors here need to be explicit regarding how they operationally investigated and explored mechanisms of sensory processing in mammalian brain, since this is the key aim of this review.
We have now added a short overview of sensory processing in C. elegans and have made a comparison with that in mammals.
- Introduction: In the first-time the authors use the abbreviation ‘C. elegans’ in the text, they should present both the spelled-out version (Caenorhabditis elegans) and the short form.
We have now corrected this so that the spelled-out name is used the first time it appears.
- Top-down mechanisms in the multisensory integration: In this paragraph, the authors described top-down mechanisms that regulate sensory integration. Since it may be challenging to readers to fully understand that the process of multisensory integration is not uniform, what factors do affect multisensory integration, and how the mammalian brain reconstructs a multisensory world at different states, I was wondering whether the author could be more specific and provide a brief overview of top-down factors that can modulate sensory processing and discuss their potential roles in multisensory integration, including any reference to support their statements.
In general, top-down factors including stress, attention, expectation, emotion, and motivation control the degree of distraction in neural processing. Since we are not the experts in this field, we are afraid we are not able to provide a more specific description. Nonetheless, we have included some excellent reviews in this regard (reference 89-92).
- Discussion: In my opinion, this review would be more compelling and useful to a broad readership if the authors moved beyond and discussed theoretical and methodological avenues in need of refinement, using this evidence to suggest a path forward. In this regard, I believe that it would have been essential to explore to use C. elegans as a model organism to study neurodegenerative diseases. In this regard, I would suggest adding evidence from recent studies that investigated hallmarks of neurodegeneration, like aggregates called “Lewy bodies” and the loss of dopaminergic neurons in frontal lobes (doi:10.17219/acem/139572; doi:10.1111/psyp.14122), which showed how overexpressing wild-type and mutated α-synuclein in dopaminergic neurons in these worms resulted in dopaminergic neuron loss. Also, in my opinion, it would be interesting to consider how human genes responsible for a range of mitochondrial diseases have an orthologous gene in preclinical models such as C. elegans, giving the possibility of using C. elegans as a model organism for studying mitochondrial diseases (https://doi.org/10.3390/cells11162607).
We have now added a paragraph describing the role of neuromodulators, including kynurenic acid, in multisensory integration. And we have included all the references mentioned.
- In my opinion, I think the ‘Conclusions’ paragraph would benefit from some thoughtful as well as in-depth considerations by the authors, because as it stands, it is very descriptive but not enough theoretical as a discussion should be. The authors should make an effort, trying to explain the theoretical implication as well as the translational application of their research, stating the potential of this review, the goal, the challenge, the knowledge or the technology necessary to achieve this goal, and future research direction, among others.
We have now rewritten the conclusion part to include some in-depth thoughts.
- In according to the previous comment, I would ask the authors to include a ‘Limitations and future directions’ section before the end of the manuscript, in which authors can describe in detail and report all the technical issues brought to the surface.
Although we agree with the reviewer that a separate section of limitations and future directions would be good to be included, descriptions of limitations and future directions are actually spread out throughout the manuscript. When we make a comparison between C. elegans and mammals, we have mentioned many limitations, including simplicity, technical restrictions, lack of complex behaviors, and so on. In addition, we have alluded to in the conclusion that we think whole brain imaging could be one of the future directions.
- Figure: According to the Journal’s guidelines, please provide a short explanatory caption for Figure 1.
A figure legend has been provided for Figure 1.
- References: I suggest presenting more references to present readers the solid background of this review manuscript. A review paper like this typically needs more than 150 references.
Overall, the manuscript contains 1 figure and 63 references. I believe that the manuscript might carry important value describing how studies using C. elegans have generated important insights into the understanding of sensory processing, including multisensory integration.
We have now added a lot more references, so that every statement has a reference(s).

Reviewer 4 Report
This is a review article about an interesting topic: multisensory integration in C. elegans in comparison to mammals.
It is a well structured manuscript. English language and style are generally fine, needing only minor editing. For example, in the title the term “with comparison” should be replaced by “in comparison”.
The introduction section describes the background of the current review in a fairly comprehensive manner.
To my opinion, the authors should add a paragraph, describing the method they used for their literature review.
Results are interesting and well presented.
Conclusion section could perhaps be rewritten in a more critical manner avoiding possible overemphasis on the value of this review.
Author Response
- It is a well structured manuscript. English language and style are generally fine, needing only minor editing. For example, in the title the term “with comparison” should be replaced by “in comparison”.
Thank you for pointing out the language issue. We have changed the title to “Multisensory Integration in Caenorhabditis elegans in Comparison to Mammals”. In addition, we have streamlined the entire manuscript and made significant revisions in a number of places.
- The introduction section describes the background of the current review in a fairly comprehensive manner.
Thank you. We have now added a short overview of sensory processing in C. elegans to strengthen the background.
- To my opinion, the authors should add a paragraph, describing the method they used for their literature review.
We have addressed this issue by stating that it is a “narrative review” both in the abstract and introduction.
- Results are interesting and well presented.
Thank you.
- Conclusion section could perhaps be rewritten in a more critical manner avoiding possible overemphasis on the value of this review.
We have now rewritten the conclusion part to include some in-depth thoughts.

Round 2
Reviewer 2 Report
The authors satisfactorily addressed all my concerns in the current form of the manuscript. It is a significant improvement over the version that was previously submitted. I recommend this one for publication.
Author Response
Thank you.
Reviewer 3 Report
20 September 2022
Manuscript ID: brainsci-1918905
Type: Review
Title: ‘Multisensory Integration in C. elegans with Comparison to Mammals’ by Yu YV et al., submitted to Brain Sciences
Dear Authors,
The authors did an excellent job clarifying all the questions I have raised in my previous round of review. Currently, this paper entitled ‘Multisensory Integration in Caenorhabditis elegans in Comparison to Mammals’, is a well-written, timely piece of research that improves the understanding of how studies using C. elegans have generated important insights into the understanding of sensory processing, including multisensory integration.
Overall, this is a timely and needed work. It is well researched and nicely written, and describes in detail the use of use C. elegans as a model organism to study multiple cognitive processes.
I believe that this paper does not need a further revision, therefore the manuscript meets the Journal’s high standards for publication. I am always available for other reviews of such interesting and important articles.
Thank You for your work.
I declare no conflict of interest regarding this manuscript.
Best regards,
Reviewer
Author Response
Thank you